# Incidence of HIV infection and associated factors among female sex workers in Côte d'Ivoire, results of the ANRS 12361 PrEP-CI study using recent infection assays

Marcellin N. Nouaman[1,2]*, Valentine Becquet[3,4], Mélanie Plazy[5], Patrick A. Coffie[1,6], Clémence Zébago[7], Alice Montoyo[8], Camille Anoma[9], Serge Eholié[1,6], François Dabis[5], Joseph Larmarange[4], for the ANRS 12361 PrEP-CI Study group[¶]

1 Programme PAC-CI, CHU Treichville, Site de Recherche ANRS, Abidjan, Côte d'Ivoire, 2 Département de Santé Publique et d'odontologie légale, UFR d'Odonto-Stomatologie, Université Félix Houphouet-Boigny, Abidjan, Côte d'Ivoire, 3 Ined, Aubervilliers, France, 4 Ceped, IRD, Université de Paris, Inserm, Paris, France, 5 Bordeaux Population Health Research Center, Université de Bordeaux, Inserm, IRD, Bordeaux, France, 6 Département de Dermatologie et Infectiologie, UFR des Sciences Médicales, Université Félix Houphouet Boigny, Abidjan, Côte d'Ivoire, 7 ONG Aprosam, San Pedro, Côte d'Ivoire, 8 ANRS, Paris, France, 9 ONG Espace Confiance, Abidjan, Côte d'Ivoire

¶ The complete membership of the author group can be found in the Acknowledgments.
* marcellinnouaman@gmail.com

**Editor:** Hamid Sharifi, HIV/STI Surveillance Research Center and WHO Collaborating Center for HIV Surveillance, Institute for Future Studies in Health, Kerman University of Medical Sciences, ISLAMIC REPUBLIC OF IRAN

## Abstract

### Background

This study aimed to estimate, using an HIV Recent Infection Testing Algorithm (RITA), the HIV incidence and its associated factors among female sex workers (FSW) in Côte d'Ivoire.

### Methods

A cross-sectional study was conducted in 2016–2017 in Abidjan and San Pedro's region among FSW aged ≥ 18 years. In addition, a sociodemographic questionnaire, HIV screening was carried out by two rapid tests. In the event of a positive result, a dried blood spot sample was taken to determine, using a RITA adapted to the Ivorian context, if it was a recent HIV infection.

### Results

A total of 1000 FSW were surveyed with a median age of 25 years (interquartile range: 21–29 years). 39 (3.9%) tested positive for HIV. The incidence of HIV was estimated to be 2.3 per 100 person-years, with higher incidence rates among those 24 years old or less (3.0% vs. 1.9%), non-Ivorian FSW (3.2% vs. 1.9%) and those with the lowest education level (4.6% in FSW who never went to school vs. 2.6%). The incidence seemed to be associated with the sex work practice conditions: higher incidence among FSW whose usual price was less than 3.50$ (4.3% vs.1.0%), FSW who had a larger number of clients on the last day of work (6.1% in those with 7 clients or more vs. 1.8%), FSW who reported not always using

**Data Availability Statement:** For ethical reasons, the full survey dataset is available only upon reasonable request at https://zenodo.org/record/5948841. An analytical dataset containing only the variables required to replicate the analysis, as well as the corresponding R script, are available in Supplementary material.

**Funding:** The PrEP-CI ANRS 12361 was funded by the Bill and Melinda Gates Foundation (Investment ID: OPP1106343) and the French National Agency for AIDS and Viral Hepatitis Research (ANRS). https://www.gatesfoundation.org/ The Bill & Melinda Gates Foundation aims to reduce inequities in health by developing new tools and strategies to reduce the burden of infectious disease.

**Competing interests:** The authors have declared that no competing interests exist.

condoms with their clients (8.5% vs. 1.5%) and FSW who reported agreeing to sex without a condom in exchange for a large sum of money (10.1% vs. 1.2%).

## Conclusion

This study confirms that FSW remain highly exposed to HIV infection. Exposure to HIV is also clearly associated with certain sex-work factors and the material conditions of sex work. Efforts in the fight against HIV infection must be intensified to reduce new infections among FSW.

## Introduction

Antiretroviral therapy (ART) decreases HIV-related morbidity and mortality as well as infectiousness, resulting in a significant reduction in the risk of HIV transmission from a treated infected person to an uninfected sexual partner [1, 2]. However, while HIV care and treatment programmes have provided access to ART for an estimated 26 million people as of 2020 worldwide, ART coverage is still far from optimal, especially in most resource-limited countries, where the HIV incidence remains relatively high. In particular, Sub-Saharan Africa is disproportionately affected by the HIV epidemic and accounts for almost 70% of HIV infections worldwide. Although new infections have been reduced by 52% since the peak in 1997, 1.7 million people had been newly infected with HIV by the end of 2019 in this region [3, 4].

Key populations and their sexual partners are at particularly high risk of HIV; in 2020, they accounted for 65% of new HIV infections worldwide. Men who have sex with men (MSM) and female sex workers (FSW) are, 25 and 26 times more likely to be infected with HIV respectively, than the general population [3]. Although the high prevalence suggests a high incidence, incidence surveys remain necessary to better understand the dynamics of the epidemic among key populations. While some data are available for MSM [5, 6], few recent incidence surveys have been conducted among FSW [7].

Moreover, HIV incidence is likely to be heterogeneous among subgroups of FSW, depending on local context, type of sex work and whether or not they attend community clinics offering various services, including HIV testing, condom distribution and prevention programmes targeting risky behaviour [8]. With the implementation of large-scale combination prevention strategies, accurate tools to monitor where and among whom new HIV infections are occurring are essential to assess the impact of these strategies and improve the effectiveness of targeted prevention programs [9–13]. In particular, new prevention tools such as pre-exposure prophylaxis are recommended by the World Health Organization (WHO) for only populations with a substantial risk of infection [14].

Therefore, it is crucial to assess the incidence of HIV infections among FSW. There are several approaches to measure the occurrence of new HIV infections in a population. The gold standard is a prospective cohorts study that analyses HIV seroconversions in uninfected individuals. However, such cohorts studies are very costly and complex to conduct. This is why the most common approach in developing countries has been inference, considering trends in HIV prevalence and assumptions about mortality and the impact of ART coverage on survival [15–18]. Recently, several laboratory approaches have been developed to distinguish newly acquired HIV infections from long-term HIV infections in cross-sectional surveys [19–21]. These incident HIV detection approaches are based on the principle that the immunological response to HIV infection evolves over several months after infection, allowing the

identification of immunological biomarkers of early HIV disease that can serve as indicators of recent infection [22]. The assay-based approach involves the use of one or more serological laboratory tests that is able to classify HIV infection according to whether the infection was acquired in the recent past. Classification using one or more assays of this kind represents an HIV Recent Infection Testing Algorithm (RITA). If accurate, incidence testing can be a rapid and cost-effective approach to obtaining reliable and up-to date information about the dynamics of HIV transmission for more effective planning [23].

An HIV RITA has been developed and adapted to the Ivorian context [24]. The present paper used this HIV RITA to estimate the HIV incidence and its associated factors among FSW in two regions of Côte d'Ivoire: Abidjan and San Pedro.

## Methods

### Study setting

The ANRS 12361 PrEP-CI cross-sectional study was designed and implemented by two Ivorian community-based organisations between September 2016 and March 2017 [25]. Aprosam works within the city of San Pedro and its surrounding areas, particularly in villages associated with farming businesses (coffee and cocoa production). Espace Confiance operates in several districts of Abidjan, the economic capital of Côte d'Ivoire (Koumassi, Marcory, Treichville, Zone 4 and Port-Bouët including its beaches). These nongovernmental organisations (NGOs) provide HIV prevention and testing services directly at prostitution sites (outreach activities) and provide HIV and sexual health care services for FSW through community clinics. Recruitment of participants for this study was made possible by the Aprosam and Espace Confiance organisations' networks of peer educators and their access to the target population.

### Study population

The study's purpose was not to represent all FSW in Côte d'Ivoire but rather to represent FSW who could be reached by the two partner organisations and who would potentially benefit from PrEP in a future programme. FSW are identified by the peer educator and they work at sites that are usually visited by them for HIV prevention/screening. Almost all of the FSW work sites were visited by the peer educator. The FSWs were recruited both in prostitution sites (brothel, hotel, bar/maquis, street, beach) and in the fixed clinics dedicated to sex workers of both NGOs.

The survey was conducted among FSW aged 18 years and older, who had never been tested for HIV or who had previously tested negative for HIV, and who worked at a sex work site at the time of the survey, in the areas targeted by the two community-based NGOs.

### Sociodemographic and behavioural questionnaire

After obtaining informed written consent, a standardised questionnaire was administered by peer educators. The questionnaire collected sociodemographic data (date of birth, nationality, place of recruitment, level of education), as well as sexual practices and behaviours (such as the duration of sex work, the age at which sex work began, the place of meeting and activity with clients, whether sex work was carried out regularly, the use of condoms during sex work, the price of the pass (the price of a single sexual encounter with a client), the number of clients, the number of condoms used on the last day of activity and the number of sexual intercourse encounters for which a condom was not used during the last seven days).

## HIV screening and laboratory analysis

HIV screening was carried out by two rapid tests (Determine®, Alere and Vikia®, bioMérieux), for all surveyed FSWs, at the sex work sites. In the event of a positive result, HIV infection was confirmed by a rapid test (stat-pack®). Then, a dried blood spot (DBS) sample was taken and transported to the laboratory of the University Hospital of Tours, France, to determine the window of infection and false positive rate using a recent infection test adapted to the Ivorian context [24] and performed directly on plasma. This recent infection test made it possible to classify HIV infections into two groups: HIV infections contracted less than 6 months prior and those contracted more than 6 months prior [24]. This recent infection test developed by Barin et al. is the Less-sensitive enzyme Immunodominant assay recent infection (EIA-RI/IDE-V3). This assay uses the enzyme immunoassay technique in a 96-well plate, based on the measurement of absorbances (OD) in one well sensitised with an equimolar peptide mixture TM (cons+D), corresponding to the immunodominant epitope of gp41 (consensus sequence envi—1 group M and consensus sequence env—1 subtype D), and in another well, sensitised with a V3 peptide solution (AE), corresponding to an equimolar mixture of the consensus sequences of the V3 region of gp120 of the HIV subtypes A, B, C, D and CRF01_AE. This test is an in-house test, applicable in the HIV-NRC virology laboratory, serology sector. The test can be performed on serum or plasma, as well as serum, plasma or whole blood on blotting paper or DBS. The test uses a mathematical formula that combines the quantitative responses to gp41 antigens in each region to distinguish between recent and established infection [24].

This in-house test has been the subject of preliminary studies using sequential serum samples from HIV-infected Ivorian patients with known dates of infection (the PRECO-CI ANRS 12277 and PRIMO-CI ANRS 1220 projects) and samples from patients at different stages of the disease (the Temprano ANRS 12136 and Trivacan ANRS 1269 trials); which allowed to distinguish a recent infection ($\leq$180 days) from an established infection (>180 days) with a window of infection (0.3 years) and false positive rate (13‰) for the Ivorian population studied.

During the study, FSW diagnosed with HIV infection were referred to community clinics by peer educators for ART.

## Description of the surveyed population

Sociodemographic characteristics and sexual practices and behaviours of the surveyed participants were described according to the study setting (San Pedro or Abidjan). Since the sample was not a random sample but rather a convenience sample of women reached by the two organisations, statistical tests such as Pearson's $\chi2$ test or Fisher's exact test could not be formally used to compare differences between the populations from the two study settings. We excluded any missing data from the percentage calculations. All analyses were performed with R version R-4.2.1 software.

## Assessment of HIV incidence

For a given population and HIV subtype, a RITA has a mean RITA duration $\omega$, defined as the mean duration for which newly infected individuals in the population have had a recently acquired infection. A RITA also has a false recent rate (FRR), noted as $\varepsilon$, which is the proportion of non recent HIV infections in the population that are misclassified by the RITA as recent.

A RITA is used to estimate HIV infection by first classifying cases of HIV infection in the population as recently acquired or not, and then applying a mathematical formula to the resulting counts of recently acquired infections. The annual incidence rate $I_r$, is estimated

using the following formula:

$$I_r = \frac{R - \varepsilon P}{(1 - \varepsilon)\omega N}$$

where $N$ is the number of HIV-negative persons in the survey, $P$ the number of HIV-positive persons, $R$ is the number of persons classified as positive by the RITA, $\omega$ is the mean RITA duration in years and $\varepsilon$ is the FRR of the RITA [26]. Sweeting et al., in 2010 describe this mathematical approach in more detail [27].

Regarding the RITA used in this study [24], the mean RITA duration ($\omega$) was 0.3 years and the FRR of the RITA ($\varepsilon$) was 0.013.

## Factors associated with HIV incidence

HIV incidence rate and their CIs were computed for the different subgroups. Unfortunately, no comparison test or multivariate analysis is currently available for RITA data.

## Ethical aspects

All FSW were informed of the risks and benefits of participating in this study before inclusion. All FSW provided written informed consent. The National Ethics Committee for Life Sciences and Health of Côte d'Ivoire approved the research protocol (N/Ref: 057/MSHP/CNER-kp of 28 June 2016).

For ethical reasons, the full survey dataset is available only upon reasonable request at https://zenodo.org/record/5948841. An analytical dataset containing only the variables required to replicate the analysis, as well as the corresponding R script, are available in S1 Data.

## Inclusivity in global research

Additional information regarding the ethical, cultural, and scientific considerations specific to inclusivity in global research is included in the Supporting Information.

## Results

### Sociodemographic and behavioural characteristics

A total of 1000 FSW, including 400 in San Pedro and 600 in Abidjan, were surveyed. The main characteristics of the surveyed population are presented in Table 1. The median age was 25 years (interquartile range: 21–29 years). The FSW surveyed in San Pedro had a lower education level than the FSW surveyed in Abidjan. The FSW in Abidjan engaged in more sex work activity than those in San Pedro. There are differences in socio-demographic characteristics between the two cities. Indeed, Abidjan is the economic capital of the country, in full expansion compared to San Pedro where the level of poverty, literacy or education is lower. San Pedro was in the recent past the largest slum in West Africa and most of the FSWs sites are within the perimeters of this slum.

### Incidence of HIV infection

Among the surveyed FSW (those never tested or with a previous negative test result), 39 (3.9%) tested positive for HIV (6.3% in San Pedro and 2.3% in Abidjan) during the survey. Of these, seven FSW were classified as being recently infected according to the RITA (average duration of infection 113 days or 0.3 years).

**Table 1. Sociodemographic and behavioural characteristics of the surveyed FSW, ANRS 12361 PrEP-CI study, September 2016—March 2017.**

| Characteristic, n (%) | San Pedro | Abidjan | Total |
|---|---|---|---|
| | **n = 400** | **n = 600** | **n = 1000** |
| **Age in years** | | | |
| **24 years or less** | 163 (41.3) | 280 (50.3) | 443 (46.5) |
| 25 years or more | 232 (58.7) | 277 (49.7) | 509 (53.5) |
| not documented | 5 | 43 | 48 |
| **Highest level of education** | | | |
| never been to school | 115 (28.9) | 105 (17.6) | 220 (22.1) |
| primary | 163 (41.0) | 219 (36.7) | 382 (38.4) |
| secondary / university | 120 (30.2) | 273 (45.7) | 393 (39.5) |
| not documented | 2 | 3 | 5 |
| **Nationality** | | | |
| Ivorian | 312 (78.0) | 378 (63.0) | 690 (69.0) |
| other nationality | 88 (22.0) | 222 (37.0) | 310 (31.0) |
| **How many years sex work has been practised** | | | |
| 3 years or less | 250 (63.6) | 381 (63.8) | 631 (63.7) |
| 4 years or more | 143 (36.4) | 216 (36.2) | 359 (36.3) |
| not documented | 7 | 3 | 10 |
| **Usual price with clients** | | | |
| 1999 CFA (~3 €) or less | 285 (71.2) | 109 (18.2) | 394 (39.5) |
| 2000 CFA (~ 3 €) or more | 115(28.7) | 489 (81.8) | 604 (60.5) |
| not documented | 0 | 2 | 2 |
| **Recruitment site** | | | |
| bar / "maquis" | 135 (33.8) | 254 (42.3) | 389 (38.9) |
| brothel | 111 (27.8) | 142 (23.7) | 253 (25.3) |
| hotel | 101 (25.2) | 58 (9.7) | 159 (15.9) |
| street | 15 (3.8) | 48 (8.0) | 63 (6.3) |
| other | 38 (9.5) | 98 (16.3) | 136 (13.6) |
| **Number of clients during last day of sex work** | | | |
| 1 client | 21 (5.2) | 154 (25.8) | 175 (17.6) |
| 2 to 6 clients | 316 (79.0) | 394 (66.0) | 710 (71.2) |
| 7 clients or more | 63 (15.8) | 49 (8.2) | 112 (11.2) |
| not documented | 0 | 3 | 3 |
| **Condom use with clients** | | | |
| never / sometimes / often | 78 (20.4) | 46 (7.9) | 124 (12.9) |
| always | 304 (79.6) | 533 (92.1) | 837 (87.1) |
| not documented | 18 | 21 | 39 |
| **Acceptance of condomless sexual intercourse in exchange for a large sum of money** | | | |
| never | 251 (69.0) | 513 (88.0) | 764 (80.7) |
| Sometimes | 52 (14.3) | 27 (4.6) | 79 (8.3) |
| often / always | 61 (16.8) | 43 (7.4) | 104 (11.0) |
| not documented | 36 | 17 | 53 |
| **Self-reported STI in the last 12 months,** | | | |
| none | 79 (20.2) | 262 (44.5) | 341 (34.8) |
| yes, at least one | 312 (79.8) | 327 (55.5) | 639 (65.2) |
| not documented | 9 | 11 | 20 |
| **Practised sex work in more than one city** | | | |
| no, 1 city only | 200 (50.3) | 527 (88.3) | 727 (73.1) |

(*Continued*)

**Table 1.** (Continued)

| Characteristic, n (%) | San Pedro | Abidjan | Total |
|---|---|---|---|
| | **n = 400** | **n = 600** | **n = 1000** |
| yes, 2 cities or more | 198 (49.7) | 70 (11.7) | 268 (26.9) |
| not documented | 2 | 3 | 5 |
| **Last medical visit with a doctor or a nurse** | | | |
| less than a year | 303 (76.1) | 367 (61.4) | 670 (67.3) |
| more than a year or never consulted | 95 (23.9) | 231 (38.6) | 326 (32.7) |
| not documented | 2 | 2 | 4 |

STI: sexually transmitted infection

The incidence of HIV was estimated to be 2.3 per 100 person-years overall (Table 2), with 3.3% in San Pedro and 1.6% in Abidjan.

## Associated factors

Some trends emerged from the results presented in Table 2.

There were variations according to the sociodemographic characteristics of the participants, with higher incidence rates among the youngest age group (3.0% in FSW 24 years old or less vs. 1.9% in FSW 25 years old or more), non-Ivorian FSW (3.2% in non-Ivorian FSW vs. 1.9% in Ivoirian FSW) and the group with the lowest education level (4.6% in FSW who never went to school vs. 2.6% in those with a primary education level and 0.8% in those with a secondary or university education level).

The incidence of HIV also seemed to be associated with the sex work practice conditions. Indeed, FSW who charged a lower price for sexual intercourse had a higher HIV exposure rate (4.3% in FSW whose usual price was less than 3.5$ vs. 1.0% in FSW whose usual price was more than 3.5$). In addition, FSW working in brothels (4.0%), in the streets (5.4%), and in hotels (4.2%) were more likely to be recently infected than those working in bars or "maquis" (0.8%). The incidence was higher among FSW who had a larger number of clients on the last day of work (6.1% in those with 7 clients or more vs. 1.8% in those with 2–6 clients) and who worked in more than one city (3.7% vs. 1.8%). The incidence did not seem to differ according to tenure in the sex industry (2.7% in FSW who performed sex work for 3 years or more vs. 2.0% in FSW who performed sex work for 3 years or less).

A higher incidence was observed among FSW who reported not always using condoms when engaging with their clients (8.5% vs. 1.5%); who reported agreeing to sex without a condom in exchange for a large sum of money (10.1% vs. 1.2); who reported contracting a sexually transmitted infection (STI) in the last 12 months (2.6% vs. 1.9%); and who had consulted a health professional more than one year previously (4.1% vs. 1.4%).

## Discussion

Our results confirm that FSW remain at high risk of exposure to HIV population in Côte d'Ivoire, with an estimated overall incidence of 2.3% (1.6% in Abidjan and 3.2% in the San Pedro region). In Côte d'Ivoire, the HIV incidence among women in the general population aged 15–64 years was estimated to be 0.03%. Specifically, it was 0.04% among women aged 15–24 years and 0.05% among those aged 25–34 years, according to the Côte d'Ivoire Population-Based HIV Impact Assessments (CIPHIA) 2017–2018 survey [28].

**Table 2. Estimated HIV incidence by HIV exposure factors among FSW, ANRS 12361 PrEP-CI study, September 2016 –March 2017.**

| | R | P | N | Estimated incidence per 100 person-years |
|---|---|---|---|---|
| **Overall population** | 7 | 39 | 961 | 2.3 |
| **Region** | | | | |
| San Pedro | 4 | 25 | 375 | 3.3 |
| Abidjan | 3 | 14 | 586 | 1.6 |
| **Age group** | | | | |
| 24 years or less | 4 | 16 | 427 | 3.0 |
| 25 years or more | 3 | 22 | 487 | 1.9 |
| **Highest level of education** | | | | |
| never been to school | 3 | 13 | 207 | 4.6 |
| primary | 3 | 17 | 365 | 2.6 |
| secondary / university | 1 | 09 | 384 | 0.8 |
| **Nationality** | | | | |
| Ivoirian | 4 | 26 | 664 | 1.9 |
| other nationality | 3 | 13 | 297 | 3.2 |
| **How many years sex work has been practised** | | | | |
| 3 years or less | 4 | 20 | 611 | 2.1 |
| 4 years or more | 3 | 19 | 340 | 2.7 |
| **Usual price with clients** | | | | |
| 1999 CFA (~3 €) or less | 5 | 24 | 370 | 4.3 |
| 2000 CFA (~3 €) or more | 2 | 15 | 589 | 1.0 |
| **Recruitment site** | | | | |
| bar/"maquis" | 1 | 9 | 380 | 0.8 |
| brothel | 3 | 16 | 237 | 4.0 |
| hotel | 2 | 9 | 150 | 4.2 |
| street | 1 | 2 | 61 | 5.4 |
| other | 0 | 3 | 133 | 0.0 |
| **Number of clients during last day of sex work** | | | | |
| 1 client | 1 | 5 | 170 | 1.9 |
| 2 to 6 clients | 4 | 27 | 683 | 1.8 |
| 7 clients or more | 2 | 7 | 105 | 6.1 |
| **Condom use with clients** | | | | |
| never / sometimes / often | 3 | 9 | 115 | 8.5 |
| always | 4 | 29 | 808 | 1.5 |
| **Acceptance of condomless sexual intercourse in exchange for a large sum of money** | | | | |
| never | 3 | 24 | 740 | 1.2 |
| sometimes | 1 | 7 | 72 | 4.3 |
| often / always | 3 | 7 | 97 | 10.1 |
| **Self-reported STI in the last 12 months** | | | | |
| none | 2 | 8 | 333 | 1.9 |
| yes, at least one | 5 | 30 | 609 | 2.6 |
| **Practised sex work in more than one city** | | | | |
| no, 1 city only | 4 | 22 | 705 | 1.8 |
| yes, 2 cities or more | 3 | 17 | 251 | 3.7 |
| **Last medical visit with a doctor or a nurse** | | | | |
| less than a year | 3 | 22 | 648 | 1.4 |

*(Continued)*

**Table 2.** (Continued)

|  | R | P | N | Estimated incidence per 100 person-years |
|---|---|---|---|---|
| more than a year or never consulted | 4 | 17 | 309 | 4.1 |

R: number of persons classified as RITA positive; P: number of HIV-positive persons; N: number of HIV-negative persons in the survey. Mean RITA duration (ω) of 0.3 years. False recent rate of the RITA (ε) of 0.13. STI: sexually transmitted infection.

Our results in FSW in Côte d'Ivoire are higher than those estimated in China by Wang et al. in 2012 and in Cotonou, Benin by Diabaté et al. in 2018, who reported rates of 1.4% and 1.4%, respectively [29, 30]. On the other hand, our result was lower than 3.5% estimated by Braunstein et al. in 2011 among FSW in Rwanda [31].

Our results are consistent with well-documented risk factors associated with HIV in previous studies [29, 31, 32]. The HIV incidence was higher among those who reported having contracted an STI in the past 12 months, those who reported not always using a condom when engaging in sex work, and those who admitted to agreeing to sex without a condom in exchange for a large sum of money than among their counterparts. This high incidence of HIV infection further evidences this among FSWs in San Pedro compared to those practising in Abidjan. In fact, they charged a lower pass price with a higher number of customers. They also had a lower rate of condom use and were more likely to accept sex without condoms in exchange for a large sum of money, and they reported more STIs than those in Abidjan. All these factors favour exposure to HIV infection.

Our results also highlight that the working conditions of FSW effect the risk of HIV exposure and acquisition. Women working in brothels, hotels or on the street had higher exposure rates than those working in bars and "maquis". It should be noted that sex work associated with bars/"maquis" are usually occasional; therefore, these FSW generally have fewer clients. A large number of clients on the last day of work was also associated with HIV acquisition. Thus, a larger number of clients increases the risk of HIV infection [29, 32].

Generally, a link between precariousness and HIV acquisition was demonstrated in our results. We observed higher incidences among less-educated FSW, younger FSM, FSW who charged a lower price for sex, FSW of foreign nationality (most often with greater social and administrative insecurity), and FSW who practised sex work in multiple cities than among their counterparts. Similarly, those who are located farther from health services have a higher risk, as the incidence was higher among those who reported not having consulted a health professional in the twelve months preceding the survey. These results corroborate the work of Szwarcwald et al. in 2018 and of Muldoon et al. in 2015 [33, 34].

We hypothesised before the survey that those who had recently entered the sex work market would be at higher risk because they have less knowledge about prevention and less capacity to negotiate condom use, but we did not observe any differences in exposure data according to the length of time they had been practising sex work. On the one hand, foreign FSW who are new to the country are usually supervised and educated by site managers about the need for systematic condom use. On the other hand, analysis of the questionnaires and qualitative interviews, which were conducted as part of the same survey and previously published [25], highlighted that despite a high rate of condom use and strong negotiation skills, FSW remain have a high HIV and other STIs exposure rate, as some sexual intercourses events do not involve the use of condoms. FSW' responses to the question assessing condom use might refer to 'typical use' as opposed to specific circumstances [25]. Also the difference between reported

STI cases and the proportion of condom use could be the effect of temporality. Indeed, STIs were reported in the last 12 months while responses on condom use during sex were related to current use, in the week or month before the survey. Also, FSW have difficulty negotiating condom use with their boyfriends or husbands, even when they do not know their HIV status and engage in sex with multiple sexual partners. They also willingly accept condomless sex with some regular clients whom they feel they can trust or when they are in high need of money.

Our study is one of the first to estimate the incidence of HIV in FSW in Côte d'Ivoire using tests to detect recent infection. Incidence data from at-risk populations are key in designing better programmes and interventions to limit new infections. Recent infection surveillance is a powerful tool with which Cote d'Ivoire's national HIV/AIDS program may identify geographic areas and demographic groups within which HIV transmission is ongoing. More broadly, similar explorations would lend important insight into transmission dynamics in a high-stigma environment. Another strength of this study is the recruitment, through peer educators, of FSW with diverse profiles from different locations.

Yet, we have to acknowledge some limitations. First, as RITA is a biological assay, classification of infections as recent or not does not rely on self-reported information. Some people with long-standing HIV infection and on treatement may be misclassified as newly infected. However, these false recent cases are taken into account, as a false recent rate is applied when estimating incidence. Secondly, our data suffer from a lack of power due to a relative sample size with a few number of FSW recently HIV-infected. In addition, the comparison between San Pedro and Abidjan could not been done with statistical tests because the sample was not a random sample but rather a convenience sample. Also, our data came from a convenience sample. It was not appropriate to present the confidence intervals. We were not able to perform a multivariate analysis. Some of the observed associations may result from interactions between several variables. For example, more foreign FSW than Ivorian FSW work in brothels, resulting in foreign FSW having larger numbers of clients.

## Conclusion

Although community-based prevention programmes for sex workers have led to the empowerment of FSW and a high rate of male condom use in general, they are not sufficient on their own to completely eliminate the risks of HIV acquisition. This study confirms that FSW, even those who have engaged in sex work for several years, remain highly exposed to HIV infection. Exposure to HIV is also clearly associated with certain sex-work factors and the material conditions of sex work.

Efforts in the fight against HIV infection must be intensified to reduce new infections among FSW. There is a need for appropriate people-centred prevention programmes that include new prevention tools, such as pre-exposure prophylaxis, and take into account the living and working conditions of FSW.

## Supporting information

**S1 Data.**
(ZIP)

## Acknowledgments

We would like to thank all participants as well as Aprosam's and Espace Confiance's peer educators and the ANRS 12361 PrEP-CI study group: Aboubakar Sangaré (Aprosam, San Pedro,

Côte d'Ivoire), Anglaret Xavier (PAC-CI, Abidjan, Côte d'Ivoire / Inserm, Bordeaux, France), Anoma Camille (Espace Confiance, Abidjan, Côte d'Ivoire), Barin Francis (Université François Rabelais, Tours, France), Bazin Brigitte (ANRS, Paris, France), Becquet Valentine (Ceped/ IRD, Paris, France), Dabis François (ISPED/Inserm, Bordeaux, France), Danel Christine (PAC-CI, Abidjan, Côte d'Ivoire / Inserm, Bordeaux, France), Eholie Serge (PAC-CI, Abidjan, Côte d'Ivoire), Ekouevi Didier (PAC-CI, Abidjan, Côte d'Ivoire), Fonsart Julien (Hôpital Saint-Louis, Paris, France), Gbosi Kate (Aprosam, San Pedro, Côte d'Ivoire), Kwamé Abo (Programme National de Lutte contre le Sida, Côte d'Ivoire), Larmarange Joseph (Ceped/IRD, Paris, France), Masumbuko Jean-Marie (PAC-CI, Abidjan, Côte d'Ivoire), Méda Nicolas (Centre Muraz, Bobo-Dioulasso, Burkina Faso), Moh Raoul (PAC-CI, Abidjan, Côte d'Ivoire), Molina Jean-Michel (Hôpital Saint-Louis, Paris, France), N'dri-Yoman Thérèse (PAC-CI, Abidjan, Côte d'Ivoire), Nouaman Marcellin (PAC-CI, Abidjan, Côte d'Ivoire), Plazy Mélanie (ISPED / Inserm, Bordeaux, France), Soh Kouamé (Aprosam, San Pedro, Côte d'Ivoire), Tanoe Solange (Espace Confiance, Abidjan, Côte d'Ivoire), Yeo Roselyne (Espace Confiance, Abidjan, Côte d'Ivoire).

## Author Contributions

**Conceptualization:** Serge Eholié, François Dabis, Joseph Larmarange.

**Data curation:** Marcellin N. Nouaman, Valentine Becquet, Joseph Larmarange.

**Formal analysis:** Joseph Larmarange.

**Funding acquisition:** Alice Montoyo, Serge Eholié, François Dabis, Joseph Larmarange.

**Investigation:** Camille Anoma, Serge Eholié, Joseph Larmarange.

**Methodology:** Valentine Becquet, Mélanie Plazy, Serge Eholié, Joseph Larmarange.

**Project administration:** Marcellin N. Nouaman, Mélanie Plazy, François Dabis, Joseph Larmarange.

**Supervision:** Marcellin N. Nouaman, Clémence Zébago, Alice Montoyo, Camille Anoma, Joseph Larmarange.

**Validation:** Serge Eholié, François Dabis, Joseph Larmarange.

**Visualization:** Joseph Larmarange.

**Writing – original draft:** Marcellin N. Nouaman.

**Writing – review & editing:** Marcellin N. Nouaman, Valentine Becquet, Mélanie Plazy, Patrick A. Coffie, Serge Eholié, François Dabis, Joseph Larmarange.

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
