## [Decision Letter · Decision Letter 0]

8 Mar 2022

PONE-D-22-02895Incidence of HIV infection and associated factors among female sex workers in Côte d’Ivoire, results of the ANRS 12361 PrEP-CI study using recent infection assaysPLOS ONE

Dear Dr. Nouaman,

Thank you for submitting your manuscript to PLOS ONE. After careful consideration, we feel that it has merit but does not fully meet PLOS ONE’s publication criteria as it currently stands. Therefore, we invite you to submit a revised version of the manuscript that addresses the points raised during the review process.

We look forward to receiving your revised manuscript.

Kind regards,

Hamid Sharifi

Academic Editor

PLOS ONE

Journal Requirements:

4. Please amend the manuscript submission data (via Edit Submission) to include author Hervé Dao.

5. One of the noted authors is a group or consortium [the ANRS 12361 PrEP-CI team]. In addition to naming the author group, please list the individual authors and affiliations within this group in the acknowledgments section of your manuscript. Please also indicate clearly a lead author for this group along with a contact email address.

Reviewers' comments:

Reviewer's Responses to Questions

**Comments to the Author**

1. Is the manuscript technically sound, and do the data support the conclusions?

Reviewer #1: Yes

Reviewer #2: Yes

Reviewer #3: Partly

2. Has the statistical analysis been performed appropriately and rigorously? 

Reviewer #1: Yes

Reviewer #2: Yes

Reviewer #3: I Don't Know

3. Have the authors made all data underlying the findings in their manuscript fully available?

Reviewer #1: No

Reviewer #2: No

Reviewer #3: Yes

4. Is the manuscript presented in an intelligible fashion and written in standard English?

Reviewer #1: Yes

Reviewer #2: Yes

Reviewer #3: Yes

5. Review Comments to the Author

Reviewer #1: This is a well written paper on the factors that influence HIV positivity in Female Sex Workers from two different cities in Côte d’Ivoire. The factors that favor HIV infection are age of less than 24 years old, being non -Ivorian, less education, if the clients have paid less, larger number of clients on a single day, inconsistent use of condoms by the client, or condomless sex, especially if extra money has been paid for the same, and presence of STDs. They have also calculated the incidence of HIV infection per 100 person years and used RITA to determine recent HIV infections of less than six months .

The paper is well written and has nothing new to offer in terms of determination of factors favoring HIV infection in FSW. However RITA is the new factor mentioned in this paper.

Some clarifications:

1. Did all FSW have both rapid tests or only one as a screening test. This is not clear in line 174.

2. In Table 1, under duration of sex activity should it not be less than 3 years or more than 3 years . It says less than 3 years and more than 4 years. What about those between 3and 4 years . Lines 267 to 269 state only below and above 4 years . Kindly rectify.

3.Abidjan saw more sexual activity than San Pedro. Yet, San Pedro shows shows higher infection rates of Hiv. A sentence or two should be added to explain this in discussion.

4. The fact that this was a convenience sample rather than a random sample and therefore certain statistical tests could not be done should be shifted to limitations of the study. Also the fact that multivariate analysis is not available for RITA.

Reviewer #2: This is a very interesting paper on the use of HIV Recent Infection Testing Algorithm (RITA) to measure recent HIV infection and HIV incidence in a convenience sample of FSW in two towns in Côte d'Ivoire. This study will be a great addition to the literature, especially as it informs the epidemiology of HIV among key populations. I have a few questions that I hope the authors can address to strengthen the manuscript (below):

Introduction

• Can the authors explain why RITA needs to be adapted to geographic context if it is an assay that measures immunological response to HIV infection (shouldn’t this biological response not be dependent on geographic context?)?

Methods

• I appreciate the authors specifying who is the target population for this study (i.e., not all FSW). However, the authors specify that part of the target population are FSW who could potentially benefit from PrEP. Which FSW do the authors believe would not potentially benefit from PrEP in the future, given the high prevalence of HIV among FSW?

• Specifically, how was recruitment done for this study? Were FSW incentivized to participate?

• Because, as the authors admit, this is a convenience sample, I do not think it makes sense to include CIs for the HIV incidence rate, which assumes the data come from a probability distribution. Could the authors please comment?

• How is the HIV incidence calculation impacted by this being a convenience sample? Put another way, how is the HIV incidence calculation robust to this being a convenience and not a random (or otherwise probabilistic) sample?

Results

• Although results are taken from convenience samples and therefore we should be cautious about directly comparing the samples, differences in the socio-demographic composition of the two samples are striking. Are there differences in the socio-demographic characteristics in the towns themselves that could explain some of the differences in the FSW populations?

Reviewer #3: The manuscript is very well-written, and speaks to an extremely important topic for improving HIV programming. A few issues require addressing in order to strengthen conclusions and improve clarity:

1) Line 170: “Price of the pass” means what?

2) Lines 176-181: over the past several years, countries have been utilizing various assays for recent infection surveillance, with varying MDRI and FRR values. The specifics of the assay and algorithm used in this study need to be described in this section.

3) Lines 311-317: given that 64.7% (per Table 1) of participants reported having had an STI in the past 12 months, while 86.9% reported “always” using condoms, perhaps the authors should comment on the reliability of reported condom use

4) Lines 318-321: it warrants mentioning that recent infection surveillance is a powerful tool with which Cote d’Ivoire’s national HIV/AIDS program may identify geographic areas and demographic groups within which HIV transmission is ongoing. This study is an excellent example of that, and similar such explorations more broadly would lend important insight into transmission dynamics in a high-stigma environment.

5) Lines 322-327: if viral load data is not integrated into the RITA, this constitutes a major limitation of the study. Without VL data, reported recent infections may include individuals who did not disclose previously known HIV+ status, who retested having been on ART, and who are therefore misclassified. The possibility of misclassification has not been discussed in this paper and needs to be addressed. Moreover, quantifying re-testers (those who test positive on recency assays, but are actually virally suppressed because of ART) would be programmatically relevant in a highly stigmatized population and environment such as this one.

6. PLOS authors have the option to publish the peer review history of their article (what does this mean?). If published, this will include your full peer review and any attached files.

Reviewer #1: No

Reviewer #2: No

Reviewer #3: No

---

## [Author Response · Author response to Decision Letter 0]

25 Apr 2022

To the editors of the Journal of the PLOS ONE

Abidjan, April 25th 2022

Response-to-reviewers letter and revised manuscript entitled “Incidence of HIV infection and associated factors among female sex workers in Côte d’Ivoire, results of the ANRS 12361 PrEP-CI study using recent infection assays”

Dear Editors,

This document provides point-by-point responses to the reviewers’ comments. Their comments, reproduced in italics, are followed by our responses and, finally by the additional text and changes (in bold red) to the manuscript first submitted. We thank the reviewers for their comments which have helped improve our paper. 

Reviewer : 1

This is a well written paper on the factors that influence HIV positivity in Female Sex Workers from two different cities in Côte d’Ivoire. The factors that favor HIV infection are age of less than 24 years old, being non -Ivorian, less education, if the clients have paid less, larger number of clients on a single day, inconsistent use of condoms by the client, or condomless sex, especially if extra money has been paid for the same, and presence of STDs. They have also calculated the incidence of HIV infection per 100 person years and used RITA to determine recent HIV infections of less than six months .

The paper is well written and has nothing new to offer in terms of determination of factors favoring HIV infection in FSW. However RITA is the new factor mentioned in this paper.

Some clarifications:

1. Did all FSW have both rapid tests or only one as à screening test. This is not clear in line 174.

2. In Table 1, under duration of sex activity should it not be less than 3 years or more than 3 years. It says less than 3 years and more than 4 years. What about those between 3and 4 years. Lines 267 to 269 state only below and above 4 years. Kindly rectify.

3. Abidjan saw more sexual activity than San Pedro. Yet, San Pedro shows shows higher infection rates of Hiv. A sentence or two should be added to explain this in discussion.

4. The fact that this was a convenience sample rather than a random sample and therefore certain statistical tests could not be done should be shifted to limitations of the study. Also the fact that multivariate analysis is not available for RITA.

Authors’ response:

1. All study participants were tested for HIV by two rapid tests. A clarification was made in the text as follows: « HIV screening was carried out by two rapid tests (Determine®, Alere and Vikia®, bioMérieux), for all surveyed FSWs, at the sex work sites » (line 179)

2. The remark is correct. We have harmonised in the tables and in the text as follows: « >3 years or more » Vs. « 3 years or less » (table 1, 2 and lines 280-281)

3. Yes, indeed the reviewer is right. Female sex workers (FSWs) in San Pedro were more likely to be exposed to HIV than those in Abidjan. Indeed, they charged a low price for the pass, which 

results in a high number of customers. They also had a lower rate of condom use and were more likely to accept sex without condoms in exchange for a large sum of money. They also reported more STIs than those in Abidjan. All these factors favour exposure to HIV infection.

The discussion section has added some elements to this effect: « The high incidence of HIV infection further evidences this among FSWs in San Pedro compared to those practising in Abidjan. In fact, they charged a lower pass price with a higher number of customers. They also had a lower rate of condom use and were more likely to accept sex without condoms in exchange for a large sum of money, and they reported more STIs than those in Abidjan. All these factors favour exposure to HIV infection ». (lines 302-307).

4. The remark is relevant. And we have completed the limitations paragraph of the study as suggested : « Yet, we have to acknowledge some limitations. Fist, as RITA is a biological assay, classification of infections as recent or not does not rely on self-reported information. Some people with long-standing HIV infection and on treatment may be misclassified as newly infected. However, these false recent cases are taken into account, as a false recent rate is applied when estimating incidence. Secondly our data suffer from a lack of power due to a relative sample size with a few number of FSW recently HIV-infected. In addition, the comparison between San Pedro and Abidjan could not been done with statistical tests because the sample was not a random sample but rather a convenience sample». Also, we have already stated in the limitations that multivariate analyses were not available for RITA. (lines 348-356)

Reviewer: 2

This is a very interesting paper on the use of HIV Recent Infection Testing Algorithm (RITA) to measure recent HIV infection and HIV incidence in a convenience sample of FSW in two towns in Côte d'Ivoire. This study will be a great addition to the literature, especially as it informs the epidemiology of HIV among key populations. I have a few questions that I hope the authors can address to strengthen the manuscript (below):

Introduction

Can the authors explain why RITA needs to be adapted to geographic context if it is an assay that measures immunological response to HIV infection (shouldn’t this biological response not be dependent on geographic context?)?

Authors’ response:

The estimation of HIV incidence by RITA depends on two key parameters : the average duration of infection (ω) and the false recent rate (ε) and their associated uncertainties, which also quantify the share of the recent among the infected and the precision of this quantification. These parameters are directly related to population levels of HIV prevalence, which determines the proportion of infected and uninfected ; and the incidence, which determines the share of the recently infected. HIV prevalence is different in different geographical contexts, that is why we say that the RITA test must be adapted to the geographical context, i.e. here to the Ivorian context. (lines 182, 185-186)

Methods

I appreciate the authors specifying who is the target population for this study (i.e., not all FSW). However, the authors specify that part of the target population are FSW who could potentially benefit from PrEP. Which FSW do the authors believe would not potentially benefit from PrEP in the future, given the high prevalence of HIV among FSW?

Specifically, how was recruitment done for this study? Were FSW incentivized to participate?

Because, as the authors admit, this is a convenience sample, I do not think it makes sense to include CIs for the HIV incidence rate, which assumes the data come from a probability distribution. Could the authors please comment?

How is the HIV incidence calculation impacted by this being a convenience sample ? Put another way, how is the HIV incidence calculation robust to this being a convenience and not a random (or otherwise probabilistic) sample?

 Authors’ response: 

We thank the reviewer for this comment. Indeed, our sample is not representative of the whole of FSWs. Nevertheless, in the same study, we assessed the acceptability of offering PrEP to this population. And almost all the FSWs interviewed would accept taking PrEP if it were offered to them.

The recruitment of participants for this study was made possible by the peer educator networks of two community organizations. FSW are identified by the peer educator and they work at sites that are usually visited by them for HIV prevention/screening. Almost all of the FSW work sites were visited by the peer educator. The FSWs were recruited both in prostitution sites (brothel, beaches, hotel, bar/maquis) and in the fixed clinics dedicated to sex workers of both NGOs. The recruitment involved FSWs who had never been tested for HIV or who had previously tested negative for HIV, and who were working at a sex work site at the time of the survey, in the areas targeted by the two NGOs. There was no incentive to participate in the study. All that was required was written consent to participate. (lines 160-164).

Due to the small sample size, we believe that it is important to provide a sense of the uncertainty of our estimates, in particular to see if differences between groups are relevant or not.

Considering that multivariate analysisis not possible with RITA, we believe that 95% CI constitutes a relevant indicator of the uncertainty.

We note that the assessment of HIV incidence in this key population is rare in our context, particularly in Côte d'Ivoire where the study took place. And the use of recent infection tests is a first and could provide a basis for future studies of incidence estimates in larger samples and in other populations in Côte d'Ivoire.

Results

Although results are taken from convenience samples and therefore we should be cautious about directly comparing the samples, differences in the socio-demographic composition of the two samples are striking. Are there differences in the socio-demographic characteristics in the towns themselves that could explain some of the differences in the FSW populations ?

Authors’ response: We thank the reviewer for this comment

There are differences in socio-demographic characteristics between the two cities. Indeed, Abidjan is the economic capital of the country, in full expansion compared to San Pedro where the level of poverty, literacy or education is lower. San Pedro was in the recent past the largest slum in West Africa and most of the FSWs sites are within the perimeters of this slum. (lines 242-246).

Reviewer: 3 

The manuscript is very well-written, and speaks to an extremely important topic for improving HIV programming. A few issues require addressing in order to strengthen conclusions and improve clarity.

1) Line 170: “Price of the pass” means what? 

2) Lines 176-181: over the past several years, countries have been utilizing various assays for recent infection surveillance, with varying MDRI and FRR values. The specifics of the assay and algorithm used in this study need to be described in this section.

3) Lines 311-317: given that 64.7% (per Table 1) of participants reported having had an STI in the past 12 months, while 86.9% reported “always” using condoms, perhaps the authors should comment on the reliability of reported condom use.

4) Lines 318-321: it warrants mentioning that recent infection surveillance is a powerful tool with which Cote d’Ivoire’s national HIV/AIDS program may identify geographic areas and demographic groups within which HIV transmission is ongoing. This study is an excellent example of that, and similar such explorations more broadly would lend important insight into transmission dynamics in a high-stigma environment.

5) Lines 322-327: if viral load data is not integrated into the RITA, this constitutes a major limitation of the study. Without VL data, reported recent infections may include individuals who did not disclose previously known HIV+ status, who retested having been on ART, and who are therefore misclassified. The possibility of misclassification has not been discussed in this paper and needs to be addressed. Moreover, quantifying re-testers (those who test positive on recency assays, but are actually virally suppressed because of ART) would be programmatically relevant in a highly stigmatized population and environment such as this one.

Authors’ response: We thank the reviewer for this relevant and important remark

1) « Price of the pass » mean the price of a single sexual encounter with a client. This clarification has been made in the text. (line 174)

2) In order to be more precise, we have completed the text : See section “Assessment of HIV incidence” for more details. (lines 185-186)

3) The point is well made. Indeed, the high percentage of STIs reported in the last twelve months contrasts with the proportion of systematic condom use. This could be the effect of temporality. Indeed, STIs were reported in the last 12 months while responses on condom use during sex were related to current use, in the week or month before the survey. Also, the presence of STIs could encourage and incite the FSW to systematically wear a condom during her sex work activity. The condom would be a means of preventing possible STIs.

We have completed the discussion as follows : FSW’ responses to the question assessing condom use might refer to ‘typical use’ as opposed to specific circumstances [ref 25]. Also the difference between reported STI cases and the proportion of condom use could be the effect of temporality. Indeed, STIs were reported in the last 12 months while responses on condom use during sex were related to current use, in the week or month before the survey. (lines 331-335)

4) We have included this remark in the manuscript: “Recent infection surveillance is a powerful tool with which Cote d’Ivoire’s national HIV/AIDS program may identify geographic areas and demographic groups within which HIV transmission is ongoing. More broadly, similar explorations would lend important insight into transmission dynamics in a high-stigma environment » (lines 342-346)

5) Viral load data are taken into account by RITA tests. Indeed, there is substantial evidence that a proportion of people with long-standing HIV infection are misclassified as newly infected by currently available tests for recent HIV infection. Therefore, the false recent rate of the RITA that depend on these tests can never be considered as zero. The false recent rate takes into account one or more of the following characteristics :

- Advanced infection, defined by a diagnosis of AIDS or a low CD4+ T cell count ;

- Ongoing antiretroviral treatment ;

- « Elite controllers » who have a low or undetectable viral load (World Health Organization, editor. When and how to use assays for recent infection to estimate HIV incidence at a population level. Geneva, Switzerland: World Health Organization; 2011)

The formula for calculating HIV incidence based on the results of a RITA therefore incorporates the false recent rate of the RITA into the incidence calculation. It is necessary to ensure that the false recent rate applied is relevant to the RITA and to the population for which the impact is estimated. This is why in our study we used the false recent rate from a sample of the Ivorian population.

A sentence was added in the discussion : « As RITA is a biological assay, classification of infections as recent or not does not rely on self-reported information. Some people with long-standing HIV infection and on treatment may be misclassified as newly infected. However, these false recent cases are taken into account, as a false recent rate is applied when estimating incidence”. (lines 348-352)

---

## [Decision Letter · Decision Letter 1]

26 May 2022

PONE-D-22-02895R1Incidence of HIV infection and associated factors among female sex workers in Côte d’Ivoire, results of the ANRS 12361 PrEP-CI study using recent infection assaysPLOS ONE

Dear Dr. N Marcellin Nouaman

Thank you for submitting your manuscript to PLOS ONE. After careful consideration, we feel that it has merit but does not fully meet PLOS ONE’s publication criteria as it currently stands. Therefore, we invite you to submit a revised version of the manuscript that addresses the points raised during the review process.

Dear Authors

Thanks so much for submitting the revised file to PLOS ONE.

The reviewers reviewed the revised file and they put a few more comments. Please revise the file based on the reviewers' comments.

We look forward to receiving your revised manuscript.

Kind regards,

Hamid Sharifi

Academic Editor

PLOS ONE

Journal Requirements:

Reviewers' comments:

Reviewer's Responses to Questions

**Comments to the Author**

1. If the authors have adequately addressed your comments raised in a previous round of review and you feel that this manuscript is now acceptable for publication, you may indicate that here to bypass the “Comments to the Author” section, enter your conflict of interest statement in the “Confidential to Editor” section, and submit your "Accept" recommendation.

Reviewer #1: All comments have been addressed

Reviewer #2: (No Response)

Reviewer #3: (No Response)

2. Is the manuscript technically sound, and do the data support the conclusions?

Reviewer #1: Yes

Reviewer #2: Yes

Reviewer #3: Yes

3. Has the statistical analysis been performed appropriately and rigorously? 

Reviewer #1: I Don't Know

Reviewer #2: Yes

Reviewer #3: I Don't Know

4. Have the authors made all data underlying the findings in their manuscript fully available?

Reviewer #1: No

Reviewer #2: Yes

Reviewer #3: Yes

5. Is the manuscript presented in an intelligible fashion and written in standard English?

Reviewer #1: No

Reviewer #2: Yes

Reviewer #3: Yes

6. Review Comments to the Author

Reviewer #1: The data availability has restricted access and is not freely available .

the manuscript needs to be edited - Table 1 and Table 2 -should state - 3 years or less and more than 3 years . no need to put more symbol as well as as state more. lines 158, 303, 308, 330, 348 need to be corrected

Reviewer #2: The authors have been responsive to reviewer comments made by myself and the other reviewers. I am mostly satisfied, however, I still strongly believe that it is inappropriate to provide confidence intervals for the HIV incidence rates because the data they are using comes from a convenience sample (not a probability-based sample). Formulas to calculate confidence intervals assume some kind of probability-based sample. I appreciate that the authors want to communicate a sense of uncertainty in their estimates but what is the benefit if the confidence intervals themselves are wrong and therefore ultimately uninformative? At the very least, this must be acknowledged in the limitations; although I think it would be more appropriate to remove the confidence intervals altogether (and explain why they are not included) or do a bootstrapping technique to estimate confidence intervals.

Reviewer #3: Thank you to the authors, once again, for this excellent manuscript. One of the initial questions was not addressed:

2)Lines 176-181: over the past several years, countries have been utilizing various

assays for recent infection surveillance, with varying MDRI and FRR values. The

specifics of the assay and algorithm used in this study need to be described in this

section.

In response to this question, the authors' response was:

2)In order to be more precise, we have completed the text : See section “Assessment

of HIV incidence” for more details. (lines 185-186)

However, the question was about the specifics of the recency assay utilized. The manuscript states (lines 179-182): "Then, a dried blood spot (DBS) sample was taken and transported to the laboratory of the University Hospital of Tours, France, to determine the window of infection (0.3 years) and false positive rate (13%) using a recent infection test adapted to the Ivorian context [24] and performed directly on plasma."

Are there no additional details available regarding this assay? What is "a recent infection test adapted to the Ivorian context?" Is it a LAg-EIA? There are at least 10 different HIV recency assays currently and commercially available. Can the authors not shed any additional light on the assay that was utilized at University Hospital of Tours, akin to the specifics provided for the initial HIV rapid testing?

7. PLOS authors have the option to publish the peer review history of their article (what does this mean?). If published, this will include your full peer review and any attached files.

Reviewer #1: No

Reviewer #2: No

Reviewer #3: No

---

## [Author Response · Author response to Decision Letter 1]

4 Jul 2022

To the editors of the Journal of the PLOS ONE

Abidjan, June 28th 2022

Response-to-reviewers letter and revised manuscript entitled “Incidence of HIV infection and associated factors among female sex workers in Côte d’Ivoire, results of the ANRS 12361 PrEP-CI study using recent infection assays”

Dear Editors,

This document provides point-by-point responses to the reviewers’ comments. Their comments, reproduced in italics, are followed by our responses and, finally by the additional text and changes to the manuscript second submitted. We thank the reviewers for their comments which have helped improve our paper. 

Reviewer : 1

The data availability has restricted access and is not freely available.

The manuscript needs to be edited - Table 1 and Table 2 -should state - 3 years or less and more than 3 years . No need to put more symbol as well as as state more. lines 158, 303, 308, 330, 348 need to be corrected

Authors’ response:

We thank the reviewer for this comment. The data are available on Zenodo.org, with the following link https://zenodo.org/record/5948841. Access to the data is available on request on the website Zenodo.org. We mentioned this in the manuscript (line 212). In addition, we have attached in this new revision of the manuscript, the scripts and the analysis reports.

Also, we have corrected tables 1 and 2 by removing the symbol at the level of 3 years or less and more than 3 which was unnecessary. Corrections have also been made to the lines 286, 287, 297, 300.

Reviewer : 2

The authors have been responsive to reviewer comments made by myself and the other reviewers. I am mostly satisfied, however, I still strongly believe that it is inappropriate to provide confidence intervals for the HIV incidence rates because the data they are using comes from a convenience sample (not a probability-based sample). Formulas to calculate confidence intervals assume some kind of probability-based sample. I appreciate that the authors want to communicate a sense of uncertainty in their estimates but what is the benefit if the confidence intervals themselves are wrong and therefore ultimately uninformative ? At the very least, this must be acknowledged in the limitations ; although I think it would be more appropriate to remove the confidence intervals altogether (and explain why they are not included) or do a bootstrapping technique to estimate confidence intervals.

Authors’ response: We thank the reviewer for this relevant and important remark. Indeed, confidence intervals provide for the HIV incidence rates are inappropriate because our data become from a convenience sample. In order to find a more appropriate solution, we used a bootstrapping technique to estimate our confidence intervals as suggested by the reviewer. These confidence intervals were very wide, so we decided to remove them on the differents estimated incidences (see table 2) and we have mentioned this in the limitations of the study (lines 374 – 375).

We have attached in this new revision of the manuscript, the scripts and the analysis reports for details.

Reviewer : 3 

Thank you to the authors, once again, for this excellent manuscript. One of the initial questions was not addressed:

2) Lines 176-181: over the past several years, countries have been utilizing various assays for recent infection surveillance, with varying MDRI and FRR values. The specifics of the assay and algorithm used in this study need to be described in this section.

In response to this question, the authors' response was:

2)In order to be more precise, we have completed the text : See section “Assessment of HIV incidence” for more details. (lines 185-186)

However, the question was about the specifics of the recency assay utilized. The manuscript states (lines 179-182): "Then, a dried blood spot (DBS) sample was taken and transported to the laboratory of the University Hospital of Tours, France, to determine the window of infection (0.3 years) and false positive rate (13%) using a recent infection test adapted to the Ivorian context [24] and performed directly on plasma."

Are there no additional details available regarding this assay ? What is "a recent infection test adapted to the Ivorian context ?" Is it a LAg-EIA ? There are at least 10 different HIV recency assays currently and commercially available. Can the authors not shed any additional light on the assay that was utilized at University Hospital of Tours, akin to the specifics provided for the initial HIV rapid testing?

Authors’ response: We thank the reviewer for this relevant and important remark

We have given further details by describing the principle of carrying out this test as follows :

This recent infection test developed by Barin et al. is the Less-sensitive enzyme Immunodominant assay recent infection (EIA-RI/IDE-V3). This assay uses the enzyme immunoassay technique in a 96-well plate, based on the measurement of absorbances (OD) in one well sensitised with an equimolar peptide mixture TM (cons+D), corresponding to the immunodominant epitope of gp41 (consensus sequence envi - 1 group M and consensus sequence env - 1 subtype D), and in another well, sensitised with a V3 peptide solution (AE), corresponding to an equimolar mixture of the consensus sequences of the V3 region of gp120 of the HIV subtypes A, B, C, D and CRF01_AE. This test is an in-house test, applicable in the HIV-NRC virology laboratory, serology sector. The test can be performed on serum or plasma, as well as serum, plasma or whole blood on blotting paper or dried blood spots. The test uses a mathematical formula that combines the quantitative responses to gp41 antigens in each region to distinguish between recent and established infection.

This in-house test has been the subject of preliminary studies using sequential serum samples from HIV-infected Ivorian patients with known dates of infection (the PRECO-CI ANRS 12277 and PRIMO-CI ANRS 1220 projects) and samples from patients at different stages of the disease (the Temprano ANRS 12136 and Trivacan ANRS 1269 trials) ; which allowed to distinguish a recent infection (≤180 days) from an established infection (>180 days) with a window of infection (0.3 years) and false positive rate (13‰) for the Ivorian population studied.

We have thus completed the paragraph “HIV screening and laboratory” in the manuscript (lines 185 - 202)

---

## [Editor Report · Decision Letter 2]

12 Jul 2022

Incidence of HIV infection and associated factors among female sex workers in Côte d’Ivoire, results of the ANRS 12361 PrEP-CI study using recent infection assays

PONE-D-22-02895R2

Dear Dr. N Marcellin Nouaman

We’re pleased to inform you that your manuscript has been judged scientifically suitable for publication and will be formally accepted for publication once it meets all outstanding technical requirements.

Kind regards,

Hamid Sharifi

Academic Editor

PLOS ONE
---

## [Editor Report · Acceptance letter]

8 Nov 2022

PONE-D-22-02895R2 

Incidence of HIV infection and associated factors among female sex workers in Côte d’Ivoire, results of the ANRS 12361 PrEP-CI study using recent infection assays 

Dear Dr. Nouaman:

I'm pleased to inform you that your manuscript has been deemed suitable for publication in PLOS ONE. Congratulations! Your manuscript is now with our production department. 

Kind regards, 

on behalf of

Dr. Hamid Sharifi 

Academic Editor

PLOS ONE